# The Wrong Assumptions of the Effects of Climate Change on Marine Turtle Nests with Temperature-Dependent Sex Determination

**DOI:** 10.3390/ani16010097

**Published:** 2025-12-29

**Authors:** Marc Girondot

**Affiliations:** Laboratoire Ecologie, Société, Evolution, Université Paris Saclay, Centre National de la Recherche Scientifique, AgroParisTech, 91190 Gif-sur-Yvette, France; marc.girondot@universite-paris-saclay.fr

**Keywords:** climate change, marine turtles, temperature-dependent sex determination, thermal radiation

## Abstract

Climate change, caused by the increase in greenhouse gases, is warming the Earth’s atmosphere and oceans. This rise in temperature affects species such as marine turtles, whose sex is determined by the temperature of their nests. Warmer nests produce more females, while cooler ones produce more males. Many studies have used air or sea surface temperature to estimate nest temperature and predict future sex ratios, but this method is questionable. The temperature inside the sand depends mainly on solar radiation and the physical properties of the beach, such as moisture, texture, and heat transfer processes. Air temperature influences the surface but not necessarily the deeper layers, where turtle eggs develop. In fact, higher air temperature may dry the sand, reducing heat conduction and possibly limiting warming at nest depth. Therefore, using air temperature as a proxy can give misleading results, and more accurate models should include soil heat dynamics and solar radiation.

## 1. Introduction

Contemporary climate change represents one of the most pervasive environmental challenges of the Anthropocene. Driven primarily by the anthropogenic increase in greenhouse gas (GHG) concentrations since the onset of the industrial era, the Earth’s climate system is undergoing rapid and unprecedented transformations. Global mean surface temperature has already increased by more than 1 °C above pre-industrial levels, with warming projected to continue throughout the 21st century under most emission scenarios [1].

The primary mechanism underlying this warming is the enhanced greenhouse effect, whereby increased atmospheric concentrations of carbon dioxide (CO_2_), methane (CH_4_), nitrous oxide (N_2_O), and other radiatively active gases trap outgoing longwave radiation, thereby altering the global energy balance. This process raises mean air and ocean temperatures and changes many physical parameters in the atmosphere.

All marine turtle species exhibit temperature-dependent sex determination (TSD), a form of environmental sex determination in which the sex of the offspring is determined by the temperature experienced during embryonic development [2]. In these species, there are no genetic sex chromosomes; instead, a thermosensitive period (TSP) occurs during development, during which differences in nest temperature strongly influence gonadal differentiation [3]. Typically, warmer incubation temperatures produce female-biased offspring, whereas cooler temperatures produce more males, following a type Ia TSD pattern. The pivotal temperature—the temperature that yields a balanced 1:1 sex ratio [4]—usually lies between 28 °C and 30 °C, though it varies slightly among species, populations, and clutches [5]. The transitional range of temperatures (TRT), within which both sexes are produced [4], is often narrow, making sex ratios highly sensitive to subtle thermal variations. The effect of incubation temperature on sex ratio can be studied at the scale of a single nest or a set of nests in which temperature loggers record the exact thermal conditions. These temperatures are used to estimate the sex ratio using a methodology that calculates a constant temperature equivalent (CTE) applied within the TSP of development for sex determination [6]. Estimation of the limits of the TSP must take into account the temperature-dependent rate of development, and the CTE relies on a thermal reaction norm for sexualization that links a given temperature to its capacity to feminize an embryo [7]. It should be noted that other crude proxies have been used in the literature, but their capacity to predict sex ratio is generally very low [8].

The effects of climate change on the primary sex ratio of marine turtles—defined as the sex ratio of offspring—were among the first impacts highlighted by researchers [9,10]. This concern is particularly relevant for species with temperature-dependent sex determination (TSD), for which rising temperatures pose a direct threat [11,12,13]. Consequently, risk analyses assessing the threat of global warming to sea turtle survival through its effects on TSD have relied on large-scale temporal and spatial proxies of the primary sex ratio.

Air temperature [14,15,16,17,18,19,20] and sea surface temperature (SST) [21,22,23] are correlated with nest temperatures, and this correlation has often been interpreted as a direct causal relationship. Consequently, most—if not all—studies evaluating the impacts of climate change on temperature-dependent sex determination (TSD) in turtles have used air temperature or SST as proxies for nest temperature [24]. Given that both air temperature and SST are increasing due to anthropogenic greenhouse gas emissions, a feminization of marine turtle populations with temperature-dependent sex determination is expected, ultimately increasing the risk of extinction.

This reliance on atmospheric and oceanic temperature proxies largely reflects the limited availability of long-term beach soil or nest temperature records, in contrast to the widespread and long-term accessibility of air and SST data [25]. However, the validity of this approach remains questionable.

First, I will review the basics of heat transfer in soil and then establish the link with the thermal effect of beach soil. Finally, I will discuss the relevance of air or sea surface temperature when studying marine turtle nests and provide recommendations.

## 2. Fundamental Mechanisms of Heat Transfer in Soil

Marine turtle eggs incubate at depths of 40 to 80 cm in the soil. Sunlight brings radiant energy that warms the soil surface. The top few centimeters of soil absorb energy, leading to a rapid temperature increase. Heat moves downward through conduction, which is a slow process, keeping deeper soil layers cooler. Understanding the mechanisms of heat transfer provides a basic framework for understanding how heat is distributed in the soil environment and reaches the nest environment. The description of heat movement in soil is based on three main mechanisms [26]:

Radiation: Radiation plays a crucial role at the soil surface and is less important in the soil itself. The soil surface receives radiation from the sun and the atmosphere and radiates heat to the atmosphere [27]. This exchange of radiation at the surface defines the boundary conditions, or starting rule, for heat transfer in the soil profile.

Conduction: Conduction heat transfer is an energy transfer mechanism in which the thermal energy is transferred from more energetic to less energetic neighboring particles [28]. Conduction occurs in solids, liquids, and gases. In soil, heat spreads from one particle to another. This mechanism is particularly effective when soil particles are compact and in close contact with each other.

Convection: In contrast to heat conduction, energy transport by convective heat transfer goes along with an actual material flow. This involves heat transfer through the movement of fluids, such as air, water, or gas. In soil, convection occurs mainly through the movement of water in wet soil. When water warms up, it can move, carrying heat with it. This phenomenon is more significant in porous soils that are sufficiently moist. When soil is dry, convection occurs mainly through the movement of air by the same principle.

### 2.1. Factors Influencing Soil Beach Heat Transfer

Several factors determine how efficiently heat transfers through sand. These factors can be broadly categorized into soil properties and environmental conditions.

#### 2.1.1. Soil Properties

Soil texture: The size and distribution of soil particles (sand, silt, and clay) significantly affect heat transfer. Sandy soils, with larger particles and less contact, generally conduct heat less effectively than clayey soils, which have smaller particles and greater surface area contact.

Soil moisture content: Water has a much higher thermal conductivity than air. As soil moisture increases, the thermal conductivity of the soil also increases, enhancing heat transfer. However, excessive water can also lead to evaporative cooling, which can counteract heat gain. The explanation of moisture’s role is therefore complex and context-dependent.

Soil organic matter: Organic matter acts as an insulator, reducing thermal conductivity. Soils with high organic matter content tend to warm up and cool down more slowly than mineral soils.

Soil porosity: The amount of pore space in the soil affects both conduction and convection. Pores filled with air reduce conduction, while water-filled pores enhance it and facilitate convection.

#### 2.1.2. Environmental Conditions

Solar radiation: The intensity of solar radiation is the primary driver of soil heating. Higher solar radiation leads to greater heat input into the soil surface. The significance of solar radiation is undeniable as the fundamental energy source.

Air temperature: Air temperature influences the heat exchange at the soil surface. Warmer air can transfer heat to the soil, while cooler air can draw heat away.

Wind speed: Wind affects evaporation and convective heat transfer at the soil surface. Higher wind speeds can increase evaporative cooling and alter the surface energy balance.

Vegetation cover: Vegetation can shade the soil surface, reducing solar radiation input. It also affects evapotranspiration, which influences soil moisture and heat transfer.

## 3. Interplay of Factors Changing Beach Soil Temperature

The interplay between these effects is complex; therefore, I will make some simplifications to discuss the anticipated impacts of GHG on beach sand temperature at the nest level (−60 cm). A schematic representation is shown in Figure 1. The transfer of energy within the soil by a mix of conduction and convection is particularly difficult to anticipate because all the necessary conditions to apply the heat equation are broken; for example, the system is not homogeneous (solid, gas, and liquid are present in soil), and the model is open.

First, it is important to recall that the sun’s energy is not affected by GHGs; there is little to no change in the total solar irradiance from the sun itself, which varies only on small solar-cycle timescales (≈0.1%). Year-round solar radiation at the timescale of this discussion (years) is nearly constant and brings an energy E1 to Earth. A fraction of the sun’s energy arriving at the surface of Earth will be transferred by radiation into the soil. If only this factor is considered, no soil temperature change is anticipated due to an increase in GHGs. A fraction of the energy arriving on the soil surface is reemitted as infrared (E3).

A fraction of the energy transported by infrared (E3) will be absorbed by GHG, and it will provoke the well-known increase in air temperature (T’1 > T1). As the air temperature increases with GHG concentrations, the rate of liquid water evaporation at the Earth’s surface will also increase, with two consequences: (1) Increasing air temperature will increase evaporative cooling. This effect could even be enhanced by increased convection due to more frequent winds when GHGs are higher, owing to higher energy in the atmosphere. (2) The cloud cover will increase. The cloud cover will trap a fraction of the sun’s energy, and only a fraction (E2) can cross the cloud cover to arrive at the surface of the soil (E’2 < E2). Then, the sun’s energy at the surface of the Earth will be lower when GHGs increase (E’4 < E4), but this effect is thought to be minimal [29].

Higher air temperatures resulting from higher GHGs (T’1 > T1) will increase the temperature in the first centimeters of soil by convection (air–soil exchange). As already stated, this effect will also increase evaporation, and the soil surface will be drier. Soil moisture exerts a profound influence on soil heat transfer, primarily by altering the soil’s thermal properties. Water has a significantly higher thermal conductivity (around 0.6 W m^−1^ K^−1^) compared to air (around 0.025 W m^−1^ K^−1^) and soil minerals (ranging from 1 to 4 W m^−1^ K^−1^, depending on mineral type). Replacing water in soil pores with air dramatically decreases the overall thermal conductivity of the soil. This means that dry soils conduct heat much less efficiently than wet soils. Water also has a high specific heat capacity (around 4186 J kg^−1^ K^−1^), much greater than that of dry soil minerals (around 800–1000 J kg^−1^ K^−1^). Consequently, decreasing soil moisture content decreases the volumetric heat capacity of the soil, meaning dry soils can store less heat energy than wet soils for the same temperature change. Evaporation of water from the soil surface is a significant energy sink. When water evaporates, it absorbs the latent heat of vaporization from the soil, leading to cooling. This evaporative cooling effect is particularly pronounced in moist soils and can significantly moderate soil temperatures, especially during periods of high solar radiation. The intention behind considering evaporation is to account for surface energy losses.

The number of parameters underlying the schematic representation in Figure 1 is large and would benefit from a more formal representation using a network approach. The most advanced implementation to date is provided by the R package *NicheMapR* v. 3.3.2 [30], but it still falls short of capturing the required level of complexity. Furthermore, determining exact values for most of these parameters is lacking in most real-world situations.

## 4. Discussion

Greenhouse gas-driven climate change primarily modifies the Earth’s energy balance through enhanced trapping of outgoing infrared radiation, not by significantly changing the amount of incoming solar radiation. However, local solar radiation at the surface can still vary due to associated changes in cloud patterns, humidity, and aerosols.

When investigating subsurface processes, air temperature represents a poor proxy for the thermal conditions experienced below ground. This discrepancy arises because subsurface temperature dynamics are primarily governed by the absorption and dissipation of solar radiation, as well as by the thermal properties of the soil, rather than by short-term variations in ambient air temperature. Consequently, solar radiation should be considered the principal driver when modeling or interpreting underground thermal processes, whereas the direct use of air temperature may lead to substantial inaccuracies.

Because air or sea surface temperatures poorly represent the thermal environment experienced by eggs in natural nests, projections based on these proxies may misestimate future sex ratios and incubation conditions. A more mechanistic approach that accounts for solar radiation and substrate heat dynamics is therefore needed to improve the accuracy of climate change impact assessments on TSD. There are few publications that use mechanistic thermal models to estimate nest temperatures in marine turtles, and none of them explicitly addresses the effects of climate change [23,31].

A temporal trend in cohort sex ratios has been described for green turtle rookeries in the northern Great Barrier Reef, where it has been shown that predominantly females have been produced for more than two decades (99.1% of juveniles, 99.8% of subadults, and 86.8% of adult-sized turtles) [32]. This study hypothesizes that nest temperature has changed in response to increasing air temperature and that this change may be responsible for the feminization of recent cohorts. However, it should be noted that historical changes in the sex ratio covary with cohort age. An alternative hypothesis is that demographic changes in population structure could explain the observed patterns. For example, changes in cohort sex ratios have also been documented in Mediterranean populations of loggerhead turtles. Primary, subadult, and adult sex ratios have been estimated for loggerheads in the Mediterranean Sea, revealing a gradient in sex ratio: a strong female bias at the primary sex ratio stage [33,34], a moderate female bias among subadults [35], and either no bias [36,37,38] or a male bias [39] in adults. These findings suggest that demographic or behavioral processes, which do not directly reflect primary sex ratios, can strongly influence sex ratios at the subadult and adult stages [40].

## 5. Concluding Remarks

Climate change, which is largely an ongoing process, not only raises mean air and ocean temperatures but also amplifies the frequency and intensity of extreme events such as heatwaves, droughts, and heavy precipitation. However, such changes are not expected to directly alter deep beach sand temperatures at the depth of marine turtle nests and may even have the opposite effect due to changes in moisture in the upper centimeters of beach soil or increased interception of solar radiation by cloud cover before it reaches the soil surface.

## Figures and Tables

**Figure 1 animals-16-00097-f001:**
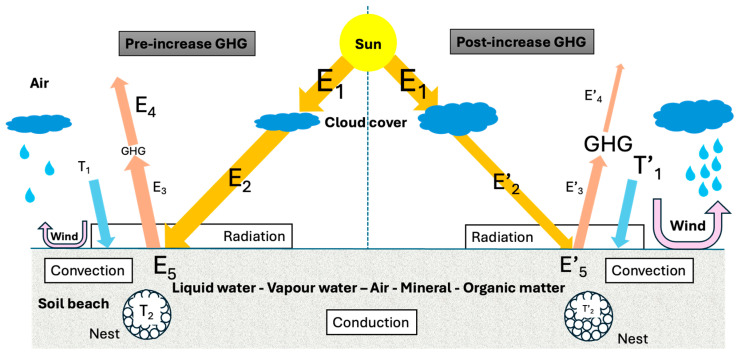
Schematic representation of the main energy transfer processes on a nesting beach before and after the increase in greenhouse gases (GHGs) due to human activity. Arrows indicate the direction of energy transfer. Droplets represent evaporation driven by rising air temperature and more wind, which leads to increased cloud cover.

## Data Availability

No new data were created or analyzed in this study. Data sharing is not applicable to this article.

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
