# Peer review of "The Wrong Assumptions of the Effects of Climate Change on Marine Turtle Nests with Temperature-Dependent Sex Determination"

_animals, 2025, doi:10.3390/ani16010097_

Round 1
Reviewer 1 Report
Comments and Suggestions for Authors
Dear Colleague,
this ms of yours sounds interesting, despite exclusively theoretical. I expect an other ms of yours with empirical data especially comparing air vs soil properties. but this is an other story.Best regards.
Author Response
this ms of yours sounds interesting, despite exclusively theoretical. I expect an other ms of yours with empirical data especially comparing air vs soil properties. but this is an other story.Best regards.
The first referee is thanked for his commentaries.
Reviewer 2 Report
Comments and Suggestions for Authors
Lines 74-80
Although the feminizing effect on the Great Barrier Reef gree turtle rookeries led the authors to hypothesize that air temperature was the main cause, the association is, in fact indirect. Nonetheless, their results from a two-decade study of greenturtles showed a clear feminization of this population. Therefore, the loation of this population is important for testing your hypothesis using the three mechanisms of heat transfer.
Lines 196-200
I fully agree that ir temperature does not accurately represents the thermal environmentexperienced by developing embryos in natural nest. Additionally, is also clear that interactions between solar radiation and the sand beach enhance the proxy for actual climate change in TSD. However, you might also consider that GHG influences TSD in natural nests depending on their geographic location. Thus, your actual hypothesis will help design predictive models based on the spatiotempral distribution of nesting populations.
Author Response
Lines 74-80
Although the feminizing effect on the Great Barrier Reef gree turtle rookeries led the authors to hypothesize that air temperature was the main cause, the association is, in fact indirect. Nonetheless, their results from a two-decade study of greenturtles showed a clear feminization of this population. Therefore, the loation of this population is important for testing your hypothesis using the three mechanisms of heat transfer.
Dear referee. Thanks a lot for yours commentaries.
I have answered in the manuscript:
A temporal trend in cohort sex ratios has been described for green turtle rookeries in the northern Great Barrier Reef, where it has been shown that predominantly females have been produced for more than two decades (99.1% of juveniles, 99.8% of subadults, and 86.8% of adult-sized turtles) [32]. This study hypothesizes that nest temperature has changed in response to increasing air temperature and that this change may be responsible for the feminization of recent cohorts. However, it should be noted that historical changes in sex ratio covary with cohort age. An alternative hypothesis is that demographic changes in population structure could explain the observed patterns. For example, changes in cohort sex ratios have also been documented in Mediterranean populations of loggerhead turtles. Primary, subadult, and adult sex ratios have been estimated for loggerheads in the Mediterranean Sea, revealing a gradient in sex ratio: a strong female bias at the primary sex ratio stage [33, 34], a moderate female bias among subadults [35], and either no bias [36-38] or a male bias [39] in adults. These findings suggest that demographic or behavioral processes, which do not directly reflect primary sex ratios, can strongly influence sex ratios at the subadult and adult stages [40].
Lines 196-200
I fully agree that ir temperature does not accurately represents the thermal environmentexperienced by developing embryos in natural nest. Additionally, is also clear that interactions between solar radiation and the sand beach enhance the proxy for actual climate change in TSD. However, you might also consider that GHG influences TSD in natural nests depending on their geographic location. Thus, your actual hypothesis will help design predictive models based on the spatiotempral distribution of nesting populations.
I do not agree at all with the referee. The GHG effects are independent on the geographic location of the beach.
Reviewer 3 Report
Comments and Suggestions for Authors
This is a Hypothesis manuscript that challenges the scientific community to move away from using air temperature as a proxy measurement for marine turtle nest thermal conditions. Instead, the author proposes integrating measures of solar radiation and soil conditions that would better represent heat dynamics and nest thermal conditions that are critical for temperature-sex determined (TSD) species.
Changes in sex ratio in response to variation in thermal profiles is a key consideration for the scientific and applied management community who focus on marine turtle species (especially given the IUCN conservation concerns). Because nest thermal conditions cannot easily be monitored without running the risk of disrupting the nesting environment (and thus violating conservation policy), proxy air temperatures are often used. This manuscript challenges the appropriateness of such proxies and presents alternatives that can be tested in future studies. Therefore, the manuscript provides a critical consideration to the conservation and scientific community.
With that said, I do have some concerns.
- There are a number of places where additional references and clearer connections are needed in Section 2 (starting on page 3). Here, the author does a nice job of outlining the key considerations for the thermodynamics of heat transfer in substrates, but given that this is critical context for your argument, there should be clearer connections to how this connects to turtle nesting based on the literature. Also, if the goal is to provide backing for your hypothesis, references that are foundational to the soil physics literature are needed.
- Along these lines, the description of conduction, convection, and radiation is presented in a fairly rudimentary way that lacks the depth expected for a scientific audience. I recommend expanding these sections as described above.
- A major issue is the context that is missing on existing methods for measuring temperatures in nests using devices such as data loggers. The author implies that proxy measurements of air surface temperatures dominate all studies, but many use data loggers in nests. I recommend acknowledging this and explaining why proxy measures are widely used (perhaps even providing data on the rough number of studies that use proxy versus other measures). Such an addition would strengthen the argument for improving proxy models.
- 1 (p.5/7) – the schematic provides a good start, but could be greatly improved. For example, include descriptions in the figure caption (or a legend) to explain the difference in red or blue and the size of the symbols. Additional recommendations:
- The lighting bots are nice for representing energy, but arrow may show the direction of energy movement more clearly and different arrow widths could represent varying intensity. Likewise, if you want to display different heat movement types (convection, radiation, conduction), you could use different line dashing (e.g., solid, dashed, dotted).
- It was unclear to me why the “convection” label was not near the T1 symbols
- The increase in evaporation due to T1 does not seem to be clearly represented.
- Continuing with Fig.1, illustrating how these processes impact the nest thermal conditions would be helpful (since that’s the main thesis of your argument). I think that’s what you’re getting at the T2, but maybe make that more obvious.
- Section 3 (p.4/7) – paragraph 3: this is an example of a place where referencing other studies would be important.
- Section 3 (p.4/7) – paragraph 4 I think your argument here could benefit from an ecological interaction network. Adding all of the variables in a formal network model could be helpful for clarity, but also it may be more familiar for your primary target audience (and more widely accessible to non-subdiscipline specialists) [see Gauzens, B., et al., (2025). Tailoring interaction network types to answer different ecological questions. Nature Reviews Biodiversity, 1-10)].
- Discussion (p.5/7) – the emphasis here (that GHG likely has no impact on nest temperature) is a very different statement than whether air surface temperatures are accurate predictors of nest conditions. This is problematic. While the author’s intent is to critique the current proxies, the phrasing deviates from that goal. I recommend explicitly stating that the hypothesis addresses the mechanistic link between air temperature and nest temperature, not the overall impact of climate change or GHG’s on marine turtle nests. The author could emphasize that the relationship between GHG and air and ocean temperatures, but clarify that the influence on subsurface nest thermal conditions is mediated by multiple interacting factors. Thus, you propose that air temperature alone is an inadequate proxy for predicting nest thermal profiles. Moreover, you may want to consider reinforcing that improving proxy models is necessary for accurate sex ration predictions under climate change scenarios rather than implying that climate change is not a threat.
Author Response
This is a Hypothesis manuscript that challenges the scientific community to move away from using air temperature as a proxy measurement for marine turtle nest thermal conditions. Instead, the author proposes integrating measures of solar radiation and soil conditions that would better represent heat dynamics and nest thermal conditions that are critical for temperature-sex determined (TSD) species.
I agree with this introduction.
Changes in sex ratio in response to variation in thermal profiles is a key consideration for the scientific and applied management community who focus on marine turtle species (especially given the IUCN conservation concerns). Because nest thermal conditions cannot easily be monitored without running the risk of disrupting the nesting environment (and thus violating conservation policy), proxy air temperatures are often used. This manuscript challenges the appropriateness of such proxies and presents alternatives that can be tested in future studies. Therefore, the manuscript provides a critical consideration to the conservation and scientific community.
I agree with this view. It is good starting point: the manuscript was understandable!
With that said, I do have some concerns.
- There are a number of places where additional references and clearer connections are needed in Section 2 (starting on page 3). Here, the author does a nice job of outlining the key considerations for the thermodynamics of heat transfer in substrates, but given that this is critical context for your argument, there should be clearer connections to how this connects to turtle nesting based on the literature. Also, if the goal is to provide backing for your hypothesis, references that are foundational to the soil physics literature are needed.
I have added on introductive sentence in this paragraph to recall that eggs are incubated at -80 to -40 cm in soil:
Marine turtle eggs incubate at -80 to -40 cm in soil.
Furthermore, I add also this information here:
Understanding the mechanisms of heat transfer provides a basic framework for understanding how heat is distributed in the soil environment and reached the nest environment.
I have added a book chapter reference:
Campbell, G. S., & Norman, J. M. (1998). Heat flow in the soil. In G. S. Campbell & J. M. Norman (Eds.), An Introduction to Environmental Biophysics (pp. 113–128). Springer. https://doi.org/10.1007/978-1-4612-1626-1_8
- Along these lines, the description of conduction, convection, and radiation is presented in a fairly rudimentary way that lacks the depth expected for a scientific audience. I recommend expanding these sections as described above.
The main idea was not to explore these phenomena in great details but to give some clues to the reader about how it works.
I have added some explanations, but I don’t want to enter in the details of equations; they can be found in many places. I prefer to stay at the phenomenology level which is more accessible for a biologist.
I have added some precisions and linked with 3 publications:
Campbell, G.S.; Norman, J.M. Heat flow in the soil. In An Introduction to Environmental Biophysics, Campbell, G.S., Norman, J.M., Eds.; Springer: New York, NY, 1998; pp. 113–128.
Bristow, R.L.; Campbell, G.S. On the relationship between incoming solar radiation and daily maximum and minimum temperature. Agricult. Forest Meteorol. 1984, 31, 159-166, doi:10.1016/0168-1923(84)90017-0.
Dincer, I.; Siddiqui, O. Heat transfer aspects of energy. Comprehensive Energy Systems 2018, 1, 422-477, doi:10.1016/B978-0-12-809597-3.00109-7.
- A major issue is the context that is missing on existing methods for measuring temperatures in nests using devices such as data loggers. The author implies that proxy measurements of air surface temperatures dominate all studies, but many use data loggers in nests. I recommend acknowledging this and explaining why proxy measures are widely used (perhaps even providing data on the rough number of studies that use proxy versus other measures). Such an addition would strengthen the argument for improving proxy models.
The first paragraph has been largely changed to make the flow of ideas clearer. I introduce earlier that temperature can be recorded within nest and can be used to predict sex ratio.
The data about northern Great Barrier Reef [Ref 32] has been moved to discussion and is now discussed in more details:
A temporal trend in cohort sex ratios has been described for green turtle rookeries in the northern Great Barrier Reef, where it has been shown that predominantly females have been produced for more than two decades (99.1% of juveniles, 99.8% of subadults, and 86.8% of adult-sized turtles) [32]. This study hypothesizes that nest temperature has changed in response to increasing air temperature and that this change may be responsible for the feminization of recent cohorts. However, it should be noted that historical changes in sex ratio covary with cohort age. An alternative hypothesis is that demographic changes in population structure could explain the observed patterns. For example, changes in cohort sex ratios have also been documented in Mediterranean populations of loggerhead turtles. Primary, subadult, and adult sex ratios have been estimated for loggerheads in the Mediterranean Sea, revealing a gradient in sex ratio: a strong female bias at the primary sex ratio stage [33, 34], a moderate female bias among subadults [35], and either no bias [36-38] or a male bias [39] in adults. These findings suggest that demographic or behavioral processes, which do not directly reflect primary sex ratios, can strongly influence sex ratios at the subadult and adult stages [40].
- 1 (p.5/7) – the schematic provides a good start, but could be greatly improved. For example, include descriptions in the figure caption (or a legend) to explain the difference in red or blue and the size of the symbols. Additional recommendations:
- The lighting bots are nice for representing energy, but arrow may show the direction of energy movement more clearly and different arrow widths could represent varying intensity. Likewise, if you want to display different heat movement types (convection, radiation, conduction), you could use different line dashing (e.g., solid, dashed, dotted).
- It was unclear to me why the “convection” label was not near the T1 symbols
- The increase in evaporation due to T1 does not seem to be clearly represented.
- The lightning bolts have been changed to arrows. The direction of arrows represents the movement of energy. However, it was not easy to include symbols for heat movement types because they are too intricated.
- The convection symbol has been moved close to T1 and Wind.
- Increase of evaporation due to both T1 and wind is shown as droplets and they are linked with cloud cover.
- Continuing with Fig.1, illustrating how these processes impact the nest thermal conditions would be helpful (since that’s the main thesis of your argument). I think that’s what you’re getting at the T2, but maybe make that more obvious.
I have added a new symbol for Nest.
- Section 3 (p.4/7) – paragraph 3: this is an example of a place where referencing other studies would be important.
Recent studies have been added in section 4 (Discussion).
Because air or sea surface temperatures poorly represent the thermal environment experienced by eggs in natural nests, projections based on these proxies may misestimate future sex ratios and incubation conditions. A more mechanistic approach that accounts for solar radiation and substrate heat dynamics is therefore needed to improve the accuracy of climate change impact assessments on TSD. There are few publications that use mechanistic thermal models to estimate nest temperatures in marine turtles, and none of them explicitly addresses the effects of climate change [23, 31].
- Section 3 (p.4/7) – paragraph 4 I think your argument here could benefit from an ecological interaction network. Adding all of the variables in a formal network model could be helpful for clarity, but also it may be more familiar for your primary target audience (and more widely accessible to non-subdiscipline specialists) [see Gauzens, B., et al., (2025). Tailoring interaction network types to answer different ecological questions. Nature Reviews Biodiversity, 1-10)].
We agree with the referee and we add this information:
The number of parameters underlying the schematic representation in Figure 1 is large and would benefit from a more formal representation using a network approach. The most advanced implementation to date is provided by the R package NicheMapR [30], but it still falls short of capturing the required level of complexity. Furthermore, determining exact values for most of these parameters is lacking in most real-world situations.
- Discussion (p.5/7) – the emphasis here (that GHG likely has no impact on nest temperature) is a very different statement than whether air surface temperatures are accurate predictors of nest conditions. This is problematic. While the author’s intent is to critique the current proxies, the phrasing deviates from that goal. I recommend explicitly stating that the hypothesis addresses the mechanistic link between air temperature and nest temperature, not the overall impact of climate change or GHG’s on marine turtle nests. The author could emphasize that the relationship between GHG and air and ocean temperatures, but clarify that the influence on subsurface nest thermal conditions is mediated by multiple interacting factors. Thus, you propose that air temperature alone is an inadequate proxy for predicting nest thermal profiles. Moreover, you may want to consider reinforcing that improving proxy models is necessary for accurate sex ration predictions under climate change scenarios rather than implying that climate change is not a threat.
I do not completely agree with referee. Let see this chain of arguments :
1/ GHG increase air temperature
2/ Heat of the nest comes mainly from solar radiation
3/ Solar radiation does not change
4/ Based on 2 and 3, nest temperature is not expected to change due to GHG
Of course the reality is more complex and it is indicated that:
The number of parameters underlying the schematic representation in Figure 1 is large and would benefit from a more formal representation using a network approach.
